# The art of compensation: How hybrid teams solve collective-risk dilemmas

Inês Terrucha[1,2]*, Elias Fernández Domingos[3,4], Francisco C. Santos[5,6], Pieter Simoens[1], Tom Lenaerts[2,3,4,7]*

1 IDLab, Ghent University-IMEC, Gent, Belgium, 2 AILab, Vrije Universiteit Brussel, Brussels, Belgium, 3 Machine Learning Group, Université Libre de Bruxelles, Brussels, Belgium, 4 FARI Institute, Université Libre de Bruxelles-Vrije Universiteit Brussel, Brussels, Belgium, 5 INESC-ID & Instituto Superior Técnico, Universidade de Lisboa, Porto Salvo, Portugal, 6 ATP-group, Porto Salvo, Portugal, 7 Center for Human-Compatible AI, UC Berkeley, Berkeley, California, United States of America

* ines.terrucha@ugent.be (IT); Tom.Lenaerts@ulb.be (TL)

**Data Availability Statement:** All computational files used to reproduce the results contained in this work are available from the Zenodo database at https://doi.org/10.5281/zenodo.10406595.

## Abstract

It is widely known how the human ability to cooperate has influenced the thriving of our species. However, as we move towards a hybrid human-machine future, it is still unclear how the introduction of artificial agents in our social interactions affect this cooperative capacity. In a one-shot collective risk dilemma, where enough members of a group must cooperate in order to avoid a collective disaster, we study the evolutionary dynamics of cooperation in a hybrid population. In our model, we consider a hybrid population composed of both adaptive and fixed behavior agents. The latter serve as proxies for the machine-like behavior of artificially intelligent agents who implement stochastic strategies previously learned offline. We observe that the adaptive individuals adjust their behavior in function of the presence of artificial agents in their groups to compensate their cooperative (or lack of thereof) efforts. We also find that risk plays a determinant role when assessing whether or not we should form hybrid teams to tackle a collective risk dilemma. When the risk of collective disaster is high, cooperation in the adaptive population falls dramatically in the presence of cooperative artificial agents. A story of compensation, rather than cooperation, where adaptive agents have to secure group success when the artificial agents are not cooperative enough, but will rather not cooperate if the others do so. On the contrary, when risk of collective disaster is low, success is highly improved while cooperation levels within the adaptive population remain the same. Artificial agents can improve the collective success of hybrid teams. However, their application requires a true risk assessment of the situation in order to actually benefit the adaptive population (i.e. the humans) in the long-term.

## Introduction

As Artificial Intelligence (AI) systems are making more and more decisions autonomously, we are relinquishing decision control, for example by allowing intelligent machines to accomplish some of our goals independently or alongside us (e.g., using Google translate to enable business opportunities across different languages [1]), within the context of hybrid human-

**Funding:** I.T., P.S. and T.L are supported by an FWO (Fond Wetenschappelijk Onderzoek) project with grant no. G054919N. E.F.D. is supported by an F.R.S-FNRS (Fonds de la Recherche Scientifique) Chargé de Recherche grant (nr. 40005955). T.L. is furthermore supported by two F.R.S.-FNRS PDR (grant numbers 31257234 and 40007793) and acknowledges also the support by TAILOR, a project funded by EU Horizon 2020 research and innovation programme under GA No 952215. E.F.D. and T.L. are supported by Service Public de Wallonie Recherche under grant n° 2010235–ariac by digitalwallonia4.ai. T.L. and P.S. acknowledge the support by the Flemish Government through the Flanders AI Research Program. The funders had no role in study design, data collection and analysis, decision to publish, or preparation of the manuscript.

**Competing interests:** The authors have declared that no competing interests exist.

machine socio-technical systems (e.g., sharing the road with self-driving cars [2]). Given the extraordinary difficulties humans have demonstrated when trying to overcome global crises, such as the COVID-19 pandemic [3] or climate change [4], the question can be raised on whether and how AI agents may help to resolve the problems in coordinating the efforts in those and similar mixed-motive situations.

Even though many different works have advocated for the introduction of beneficial AI to promote human prosociality [5–7], others have pointed out that humans may be keen to exploit this benevolent AI behavior in their own favor [8–11]. Thus, before flooding society with AI applications with the promise that they could solve some of the most pressing societal issues, it is worth asking: What behavioral responses can be expected in the presence of AI partners? How may decision-making potentially be affected? Will hybrid groups involving AI agents with predefined decision processes (and humans capable of adapting their own behavior freely) actually achieve greater collective success?

We frame here these questions within the context of the Collective Risk Dilemma (CRD) [4], a game that abstracts the conflict between helping the group to achieve a future goal at a personal cost, or free ride on the efforts of others and just collect the gains associated with achieving the goal (see Methods). The CRD is a public goods game with a delayed reward that is associated with societal problems like pandemic control through vaccination, climate negotiation to achieve $CO_2$ reduction and energy-grid usage by prosumers. Many experiments to assess human behavior have been performed [4, 12–15], where [16] found that replacing all human action by artificial delegates has a positive impact on the success rate in the CRD. They also showed that hybrid groups, i.e. groups made of both humans and artificial agents with a pre-defined, non-adaptive behavior, do not achieve higher success rates than non-hybrid (human-only) groups of the same total group size.

Here we expand on that experimental work and hypothesize that the way one evaluates the performance of the hybrid team, namely, the control group used to compare it with, will influence the resulting observations. In the experimental work carried out in [16], on the one hand, the control group is the original non-hybrid team of the same size, which by the *substitution* of some group members by artificial agents forms the hybrid team. On the other hand, one may consider also the notion of hybrid group formation via *addition*: artificial agents are introduced in non-hybrid teams without substituting any pre-existing group members. Changing the original non-hybrid team, or the context of team formation, in this manner, may have repercussions on the performance metrics of the resulting hybrid team (see Methods). This manuscript aims to unravel the behavioral dynamics of hybrid human-agent groups, and provide knowledge on how to better assess and curate human-AI teams to tackle mixed-motive and competitive scenarios of varied risk probabilities in accordance to the non-hybrid context from which they derive (*substitution* or *addition*).

Two types of participants are considered in the hybrid groups, i.e. adaptive individuals that can change their behavior over time based on the outcome of their interactions (a proxy for potential human responses) and artificial agents that capture an automated probabilistic response that does not change as a result of the interactions (a proxy for average artificial agent behavior). We apply a social learning approach (see Methods) to alter the strategy of the first type of individuals. They can switch between the possible actions in function of their success in the interactions; both when there are other adapting individuals or stochastic artificial agents in groups of a given size. Such social adaptation can be achieved in different ways (e.g. Roth-Erev learning [17], Q-learning [18]), but here an evolutionary game theoretical approach is considered wherein strategic behaviors change in the adaptive individuals population by imitating those individuals that are performing the best [19–23].

In our model, the automated response of every artificial agent does not consider continuous learning; rule-based systems are often used in AI products and they are hard-wired in the systems, as learning on the fly might be costly or even dangerous. Thinking about real-world AI applications, one should always consider that producers of AI products want to give guarantees on what the product does (also on what its limitations are, which is why we model agent behavior as a stochastic process that includes errors), and that allowing for extensive adaptation while in use may be very risky. It is also important to note that in this work, we are not considering the artificial agent designer, neither the dynamics involved behind them, whose self-interest might lead to the implementation of different behaviors in the artificial agents. We are simply probing: If we consider this space of behaviors for artificial agents, what kind of human behaviors may potentially emerge given constant hybrid interactions? Even though CRD scenarios may be used to model very high risk events like a pandemic or the climate change, the same kind of non-linearity could be observed within many industrial or software hybrid teams, where, if the project is not delivered, may suffer the consequences of losing their bonuses or even their jobs. Moreover, many teams may already could be considered hybrid if one takes into account the productivity software agents that are already used to make development or business processes more efficient.

We will show that, at least when the risk is high, the adaptive individuals in hybrid teams will exploit the benevolence of the artificial agents in their group, by avoiding to contribute with cooperative efforts when their artificial counterparts are already meeting the threshold needed, as previously hinted to in [11]. On the contrary, when the artificial agents added to each group are associated with a lower capacity to contribute for the collective endeavor, because the risk of loss is high, the adaptive population will boost its cooperation levels to compensate for the bad behavior of the agents. However, when we assess whether or not the hybrid team performs better than the original non-hybrid team, we generally find that success rates only increase in lower risk situations. The exception being when the original non-hybrid team was actually unable to reach the threshold on its own, in which case adding artificial members to its group will always increase the success rate in the long-term.

## Related work

In [8] it is pointed out that more experimental research is needed to really understand how human strategic decision-making changes when interacting with autonomous agents. Following on this [9], compiles a review of more than 90 experimental studies that have made use of computerized players. Its main conclusions validate that indeed, human behavior changes when some of the other players are artificial, and furthermore, the behavior deviates to become more rational (or in other words, selfish), where humans are observed to actually try to exploit the artificial players.

This last conclusion was both supported by [10, 11]. The first finds that humans cheat more against machines than against other humans, and thus prefer to play with machines, in an experiment that tested honesty in opposition to the possibility of higher financial gains. The latter recently published an experimental study that concludes that humans are keen on exploiting benevolent AI in various different classical social dilemma games. Within the context of the CRD used for the present work [16], groups participants in hybrid teams with AI agents. Even though 3 out of the 6 group members were AI agents that were successful in avoiding the risky outcome in previous treatments, the hybrid groups were not more successful than only-human groups. Looking closer at the results, one can see that the average payoff of the humans in hybrid teams actually increases. These experimental results already hint towards the adoption of a compensatory behavior on part of the human members of the

group once they are informed about the addition of somewhat collaborative agents to the group.

In contrast with aforementioned works [5, 6] point, towards the possibility of engineering prosociality in human behaviour through the use of pro-social AI agents. In the pursuit of this idea [7], assembles a comprehensive review on the use of robots and virtual agents to trigger pro-social behaviour. Out of 23 studies included, 52% reported positive effects in triggering such cooperative behavior. However, 22% were inconclusive and 26% reported mixed results. Moreover, while recent experimental works show that programming autonomous agents [24] that include emotion [25] or some form of communication [26] may positively impact human cooperation, it is still unclear what are the mechanisms facilitating this effect.

More directly related to our theoretical study, there are different works on the dynamics of how evolving populations adapt their behavioral profile given the introduction of agents with a fixed behavior (usually cooperative) either at the group level or at the population level [27–32]. With our research questions, we also aim at understanding how the introduction of agents with a fixed behavior, not necessarily cooperative, affects the evolution of cooperation.

## Methods

### The one-shot Collective Risk Dilemma (CRD)

In this manuscript we adopt the $N$ person one-shot CRD [18, 33–38]. Here, a group of $N$ individuals must each decide whether to Cooperate ($C$), by contributing a fraction $c$ of their initial endowment $b$, or to Defect ($D$) and contribute nothing. If the group contains at least $M$ $C$ players, i.e., the group contributes in total $Mcb$ ($M \leq N$) to the public good, then each player may keep whatever is left of their initial endowment. Otherwise, there is a probability $r$ that all players will loose all their savings and receive a payoff of 0, hence the dilemma. Thus, the expected payoff of a $D$ and a $C$ player can be defined in functions of the number of $Cs$ in the group, $j$:

$$\pi_D(j) = b(1 - r + r\theta(j - M)) \tag{1}$$

$$\pi_C(j) = \pi_D - cb \tag{2}$$

where $\theta(x)$ is the Heaviside unit step function, with $\theta(x) = 0$ if $x < 0$ and $\theta(x) = 1$ otherwise.

### CRD with hybrid interactions

We consider a population $H$ of $Z$ adaptive individuals which are randomly sampled into groups of size $N - a$ to play the CRD with $a$ artificial agents from population $A$ (whose individuals display a fixed averaged behavior). This allows us, as explained in the section below, to investigate the population dynamics of this dilemma. When engaging in group interactions, each adaptive individual can either cooperate $C$ or defect $D$. The state of the population is then defined by the number of cooperators $k \in [0, Z]$. The behavior of the fixed artificial agents is defined by their probability of cooperating in each interaction, $p \in [0, 1]$, thus, they implement a stochastic (or mixed) strategy. In each group we can calculate the expected payoff of $Ds$ or $Cs$ in function of the number of cooperators from the adaptive population, $i$, the number of fixed agents $a$ and the payoff of a D (C) $\pi_{D(C)}$:

$$\Pi_{D(C)}(i, a, p) = p\pi_{D(C)}(j = i + a) + (1 - p)\pi_{D(C)}(j = i) \tag{3}$$

The behavioral dynamics exhibited by the population of adaptive individuals are governed by a social learning mechanism, where two randomly chosen individuals compare their fitness and imitate the one who is more successful within their social environment [19–23]. Their

fitness is the measure of the success of their current strategy (their payoff) averaged over all different group interactions. It can be defined as a function of the aforementioned variables by taking into account the population state and the payoffs given by Eqs (1) and (2). Following on this, the fitness equations for cooperative (C) and defective (D) strategies, can be written as:

$$f_C = \binom{Z-1}{N-a-1}^{-1} \sum_{i=0}^{N-a-1} \binom{k-1}{i} \binom{Z-k}{N-a-1-i} \Pi_C(i+1,a,p) \tag{4}$$

$$f_D = \binom{Z-1}{N-a-1}^{-1} \sum_{i=0}^{N-a-1} \binom{k}{i} \binom{Z-k-1}{N-a-1-i} \Pi_D(i,a,p). \tag{5}$$

Each individual in the adaptive population may change its strategy profile at a given evolutionary step in the following way: an agent with a D (C) strategy is randomly selected from the population $H$ to adapt. With probability $\mu$ it will mutate into a C (D) strategy, otherwise, with probability $1 - \mu$, it will compare its fitness with another randomly selected individual (assuming the newly selected individual has a different strategy) [19–23, 33, 39]. In case imitation is selected, a D (C) strategy will turn into a C (D) with a probability

$$P(D \rightarrow C) = \frac{1}{1 + e^{-\beta(f_C - f_D)}} \tag{6}$$

described by the Fermi function. This changes the state of the population $H$ of adaptive agents from $k$ to $k + 1$. This probability becomes higher with a larger difference between the fitness of the two agents, $f_C - f_D$, or with a larger selection strength of the process, $\beta$.

The transition probabilities that regulate the stochastic dynamics of population $H$, by defining the probability of increasing (+) or decreasing (-) the number of cooperators within a population are given by:

$$T^+(k) = \frac{Z-k}{Z} \left( (1 - \mu) \frac{k}{Z-1} P(D \rightarrow C) \right) \tag{7}$$

$$T^-(k) = \frac{k}{Z} \left( (1 - \mu) \frac{Z-k}{Z-1} P(C \rightarrow D) \right) \tag{8}$$

where $P(C \rightarrow D)$ is obtained by replacing $C$ with $D$, and $D$ with $C$ in Eq (6).

From these equations, we can construct the complete Markov chain of the $Z + 1$ different states that fully describe the evolutionary process of the population $H$. From this Markov Chain we can compute the stationary distribution $P(k)$, the average cooperation level $\overline{C}$ and the average group success $\overline{s}_G$ of each population configuration.

To compute the stationary distribution $P(k)$, we retrieve the eigenvector corresponding to the eigenvalue 1 of the tridiagonal transition matrix $S = [p_{ij}]^T$ [20, 22, 23]. The values $p_{ij}$ are defined by the equations:

$$p_{k,k\pm1} = T^\pm(k) \tag{9}$$

$$p_{k,k} = 1 - p_{k,k-1} - p_{k,k+1} \tag{10}$$

where the formulas that define $T^\pm(k)$ can be consulted in Eqs (7) and (8).

From this it follows that the cooperation level $\overline{C}$ of population $H$ (for a given set of parameters $N$, $M$, $r$, $a$ and $p$) by averaging the fraction of cooperators in each population state, $k/Z$,

over the stationary distribution of states $P(k)$ as given by:

$$\overline{C} = \sum_{k=0}^{Z} P(k)\frac{k}{Z} \qquad (11)$$

As already mentioned, within the context of the CRD [4, 18, 33–38] another relevant quantity to derive is the probability of success of each group in reaching the threshold necessary of $M$ cooperators to avoid the collective risk.

At the population level, we compute the fraction of groups in each population state that are successful by resorting to the multivariate hypergeometric sampling, as follows

$$s_G(k) = \binom{Z}{N-a}^{-1} \sum_{h=0}^{N-a} \binom{k}{h}\binom{Z-k}{N-h-a} \times$$
$$\times (p\theta(h+a-M) + (1-p)\theta(h-M)) \qquad (12)$$

where $\theta(x)$ is the Heaviside unit step-function as in Eqs (1) and (2) of the main text.

Finally, similarly to what was done in Eq (11), we calculate the averaged group success by weighing the group success of each population state of Eq (12) over the stationary distribution of the evolutionary process $P(k)$:

$$\overline{s}_G = \sum_{k=0}^{Z} P(k)s_G(k) \qquad (13)$$

The code used to compute all the aforementioned quantities and reproduce the results that constitute this manuscript is made available at [40].

## Forming hybrid teams through substitution or addition

As mentioned above, hybrid teams are formed by $N - a$ adaptive individuals and $a$ artificial agents with a fixed behavior. Using Eqs (11) and (13) we are able to calculate the average cooperation level and success rate of a population $H$ that evolves by playing the CRD in hybrid teams with that configuration. To determine how well the adaptive population fares when playing in hybrid teams, we should be able to compare these metrics with the results obtained for non-hybrid teams, i.e. teams with $a = 0$, therefore only made of $N$ adaptive individuals.

To this end, we introduce the idea of a control population, corresponding to a population $H_C$ that evolves by playing the CRD in non-hybrid teams, and for which we compute both its cooperation level $\overline{C}_C$ and its success rate $\overline{s}_C$ (subscript $C$ refers to the *control*) following Eqs (11) and (13), respectively, by adjusting the parameters $N$ and $a$ according to the control configuration of its non-hybrid teams. We assume that hybrid teams are a result of either one of two different processes—*substitution* or *addition* -, so that the control chosen to evaluate the relative results of the hybrid team configuration differ, depending on the process used to form the hybrid team.

The process of *substitution* refers to the formation of a hybrid team out of a non-hybrid team of $N$ group members by substituting a number $a$ of these group members by artificial agents. In this case, the non-hybrid team used as control will have $N$ of adaptive individuals only (a total group size $N_C = N$), whereas the hybrid team whose performance we are evaluating will also have $N$ group members, where $N - a$ are adaptive individuals and $a$ are artificial agents. To calculate cooperation level and success rate for the control population with a non-hybrid team group configuration—$\overline{C}_S$ and $\overline{s}_S$ (with a subscript $S$ for substitution) -, we simply take Eqs (11) and (13) and substitute $a \to 0$, maintaining all the other variables already used to

compute $\overline{C}$ and $\overline{s}_G$ for the case of hybrid teams. In this case, we can write the relative gain in cooperation level $\Delta\overline{C}_S$ and the relative gain in success rate $\Delta\overline{s}_S$ by the process of substitution as shown below in Eqs (14) and (15), respectively.

$$\Delta\overline{C}_S = \overline{C}_G - \overline{C}_S = \overline{C}_G - \overline{C}_G(a \to 0) \tag{14}$$

$$\Delta\overline{s}_S = \overline{s}_G - \overline{s}_S = \overline{s}_G - \overline{s}_G(a \to 0) \tag{15}$$

Another way to form hybrid teams, rather than substituting group members by artificial agents, is to simply add new members to the non-hybrid team which are artificial agents—a process we define as *addition*. In this case, the non-hybrid team used as control will be made of $N_C = N - a$ adaptive individuals only, and it is to be compared with hybrid teams made of $N - a$ adaptive individuals and $a$ artificial agents, so $N = N_C + a$. We consider this case to be a more useful comparison for real applications, since from the ethical standpoint on the future of work, hybrid teams should be created as an attempt to enhance human performance, rather than to substitute human labour [41]. In this case, to calculate the cooperation level and success rate of the control population, $\overline{C}_A$ and $\overline{s}_A$ (with a subscript $A$ for addition), we use the same Eqs (11) and (13) as for the accompanying hybrid scenario but we substitute $a \to 0$ and $N \to N - a$, since for this control there are no artificial agents and the group size of the team is considered to be the same as the number of adaptive individuals that are in the hybrid team ($N_C = N - a$ in this case). For this case of *addition*, Eqs (16) and (17) show how to calculate the gain in cooperation level ($\Delta\overline{C}_A$) and in success rate ($\Delta\overline{s}_A$) obtained by the formation of hybrid teams.

$$\Delta\overline{C}_A = \overline{C}_G - \overline{C}_A = \overline{C}_G - \overline{C}_G(a \to 0, N \to N - a) \tag{16}$$

$$\Delta\overline{s}_A = \overline{s}_G - \overline{s}_A = \overline{s}_G - \overline{s}_G(a \to 0, N \to N - a) \tag{17}$$

## Results and discussion

### Compensation ensures high success, but adaptive individuals do not cooperate when artificial agents do

To probe the question of whether or not to introduce artificial agents and form hybrid teams to tackle a CRD, one of the first metrics to study is the success rate (Eq (13)) obtained by such groups. Fig 1 shows how the average group success of an evolving population of adaptive individuals fares if they were playing the CRD within hybrid teams in three conditions; i) $a < M < N - a$ (Fig 1A), ii) $a = M = N - a$ (Fig 1B) and iii) $a > M > N - a$ (Fig 1C). The difference between all three panels is in which type of individuals, i.e. adaptive (who are proxies for humans) or fixed (who are proxies for artificial agents) behaviour individuals, can achieve group success by themselves: in Fig 1A the threshold can be solely met by the adaptive ones, in Fig 1B both the adaptive and the artificial agents can reach the threshold alone, while in Fig 1C only the artificial agents can reach the threshold $M$ by themselves. In all three scenarios, artificial and adaptive agents can cooperate together to reach the threshold $M$ because $(N - a) + a = 6$ which is always bigger than $M = 3$.

Starting by Fig 1A, one can immediately distinguish between high-risk scenarios where success rate is almost always very high, and low-risk scenarios where reaching success appears to be more difficult (almost) independently of other variables. Prior work on the CRD [18, 33–38] identified this non-linearity in the game, which is related to the fact that in high-risk

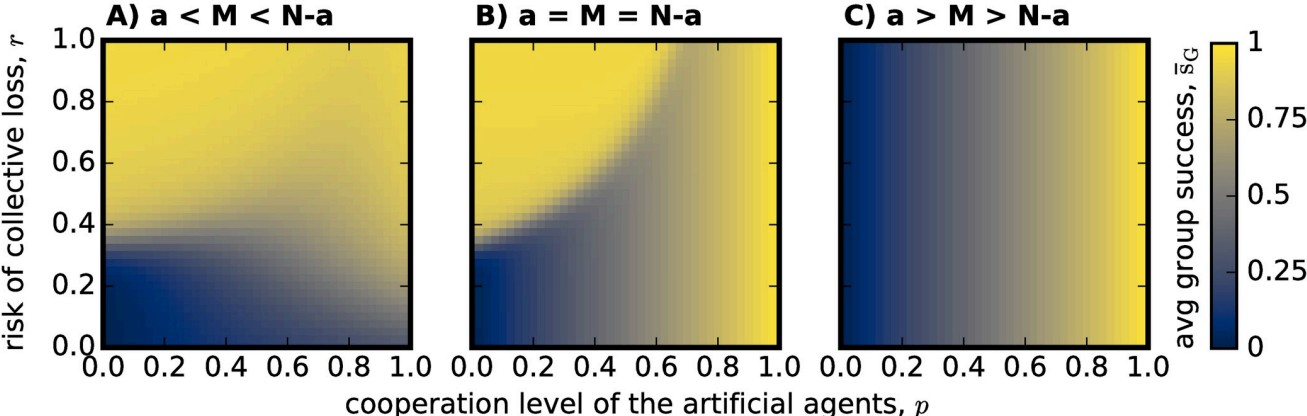

**Fig 1. Average group success obtained for an adaptive population of individuals playing a CRD in hybrid groups.** All images portray hybrid groups of group size $N = 6$ where at least $M = 3$ must be cooperators in order to avoid risking losing their endowments with risk probability $r$ in the $y$–axis. The $x$–axis shows the probability of the $a$ artificial agents in the group, who implement a fixed stochastic strategy, cooperating: $p$. Average group success is illustrated following the color bar on the left side of the figure. In (A) hybrid groups are made of $N − a = 4$ adaptive individuals and $a$ artificial agents, so that $a < M < N − a$ and adaptive individuals are able to meet the threshold on their own. In this case, for higher risk $r$, the hybrid groups achieve higher success, a trend that is fairly independent from the probability of the artificial agents cooperating $p$. In (B) $a = M = N − a = 3$, again adaptive individuals can meet the threshold $M$ on their own, but the dilemma is harder since all of them would have to cooperate in order to do so. Two regions of high success are surrounded by a shadow of lower success rates: when $p$ is very high, independently of risk probability $r$; and a semi-circle on the top left corner of the picture where risk is high and $p$ is low. (C) represents hybrid groups made of $N − a = 2$ adaptive individuals and $a = 4$ artificial agents, so that adaptive individuals would not be able to meet the threshold $M$ on their own as $a > M > N − a$. In this case, average group success appear to change only in accordance to the probability of the artificial agents cooperating $p$, independently from the risk of collective loss $r$. These results follow Eq (13), using the parameters: $Z = 100$, $\mu = 0.01$, $\beta = 2$, $b = 1$, $c = 0.1$.

situations, individual payoffs are highly dependent on collective success whereas in lower risk CRDs, it is more profitable to take the risk than to cooperate. In Fig 1A, where $a < M$ and therefore the success of the hybrid team depends on the $N − a$ adaptive participants of the group, this trend is clearly identifiable. In this case, variable $p$ (see Eq (3)), that identifies the given average level of cooperative behavior of the artificial agents in the group, appears to simply dim the risk boundary that separates high and low success rates, with the transition being very abrupt for lower $p$ but smoother for higher $p$, in which case the success rate is never very low, even for low-risk scenarios.

When $a = N − a = M$ in Fig 1B, and therefore the threshold could be met by either the artificial or the adaptive agents (or a combination of both, as in all cases), we see that indeed when $p$ is high—the artificial agents are very cooperative -, high success rates are uniformly achieved for any risk factor. This is due to the fact that cooperation by the artificial agents alone is sufficient to secure collective success for this case of $a \geq M$. However, for high risk, there is a shadow of lower success rates for decreasing $p$. A semi-circle of high success can also be observed in the top left region (high risk and low $p$). Since within this semi-circle the cooperative level of the agents $p$ would not be sufficient to ensure collective success, this semi-circle must be associated with high levels of cooperation on the part of the adaptive individuals. Yet, and this is highly intriguing, a shadow of lower success separates this semi-circle from the region where the artificial agents are cooperative enough to achieve success by themselves. When comparing that region of reduced success between the two extremes with the results in Fig 1A, the adaptive individuals seem to have become less effective in achieving the goal, even for high risk.

Finally, Fig 1C depicts a situation where the adaptive population by itself would not be able to meet the required threshold to avoid collective disaster: $N − a < M$. The figure shows how

collective success is completely dependent on the cooperation level encoded in the behavior of the artificial agents in the group, i.e. $p$. When $p$ is sufficiently high, then maximum success rates are observed, and when $p$ drops towards 0, success rate plummets accordingly.

To explain high success rates in regions of low $p$, or when $a < M$, in Fig 1, we hypothesize that the cooperation level of the adaptive population must be compensating the insufficient cooperative effort of the artificial agents in these cases. To support this hypothesis, Fig 2 shows the average cooperation level observed for the adaptive population only, following Eq (11).

In Fig 2A, where adaptive individuals can reach the threshold on their own $N - a > M$, we observe that their cooperation level alone is able to explain how the success rate changes with respect to $r$ and $p$, showing an almost fully correlated result to the one of Fig 1A. Interestingly the cooperation level for the adaptive population presented in Fig 2B can only account for one of the regions of high success previously identified in Fig 1B, given that the other one is connected with the cooperative efforts of the artificial agents. Assigning an important role to the artificial agents in the group, alters the overall behavior of the adaptive individuals: they become more cooperative in the high-risk region but restrict their effort to a specific area of the parameter space. Finally, since the adaptive population presents no cooperation whatsoever in Fig 2C, we can confirm that the success rates observed in Fig 1C are fully justified by the efforts of the artificial agents in this case. Overall, we see that success can be obtained for a wide array of different parameters ($r$ and $p$) through a mechanism of compensation developed by the adaptive population to cooperate whenever the artificial agents are not cooperating enough.

However, the same mechanism will lead adaptive individuals to fully refrain from cooperation when the number of artificial agents in the group and their cooperative effort is perceived to be sufficient to attain collective success. In a way, the compensation mechanism can then almost be described as an exploit of the artificial agents by part of the adaptive population. Indeed, a negative consequence of this evolution of the adaptive population, is that even for high risk scenarios, less than optimal success rates are obtained in Fig 1B and 1C. In Fig 1B,

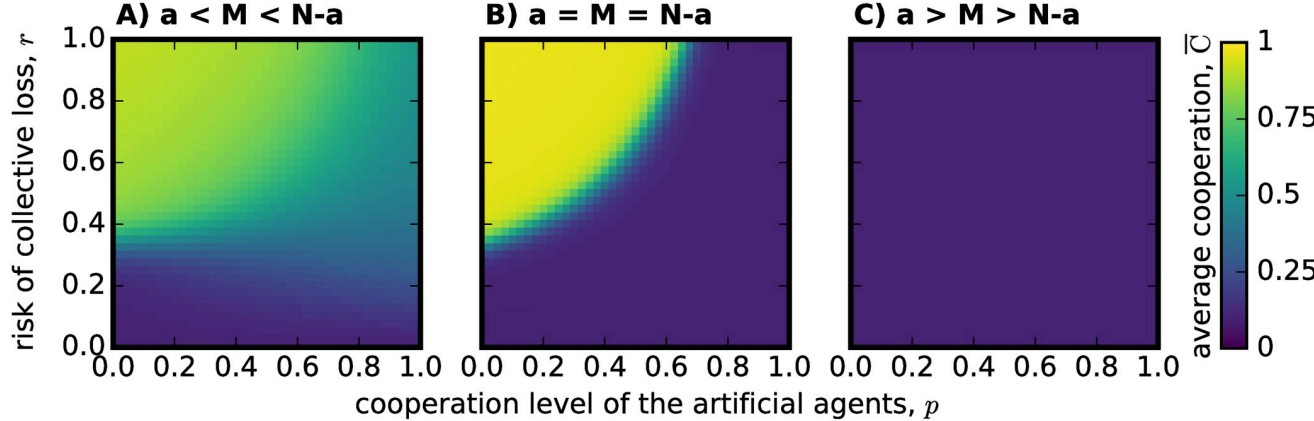

**Fig 2. Average cooperation level obtained for an adaptive population of individuals playing a CRD in hybrid groups.** The same set-up (using the same parameter choices) used in Fig 1 is recreated, but now we calculate average cooperation level following Eq (11). In (A) where $N - a = 4$ are adaptive individuals and $a$ artificial agents, so $a < M < N - a$, we see that the average level of cooperation in the adaptive population appears to solely explain the average group success observed in Fig 1A. In (B), where $a = M = N - a = 3$, adaptive individuals can again meet the threshold $M$ on their own, but the dilemma is harder than in A), since all of them would have to cooperate in order to do so. High cooperation levels associated with the adaptive population are restricted to an enclosed semi-circle on the top left of the figure, for high risk probability $r$ and for lower cooperation levels on the part of the artificial agents $p$. (C) represents hybrid groups made of $N - a = 2$ adaptive individuals and $a = 4$ artificial agents, so that adaptive individuals would not be able to meet the threshold $M$ on their own as $a > M > N - a$. In this case, the adaptive population does not cooperate at all.

the shadow of less than optimal average success rate could have been avoided if the adaptive population was to cooperate more outside of the semi-circle observed in Fig 2B; and in Fig 1C success could have extended for a wider range of *p* values if the adaptive population would have cooperated a bit to complement the efforts of less cooperative artificial agents. In both cases, a possible expectation mismatch about the artificial agents' efforts led the adaptive population to relax too much their own cooperative load.

## Hybrid teams achieve higher success rates in low-risk CRDs than teams without artificial agents

One of the main arguments for introducing artificial agents in a group to form hybrid teams with adaptive agents, is precisely to test it as a solution to enhance group success in the CRD, a question experimentally analysed in [16]. We thus analyse the two types of hybrid group formation mechanisms discussed in the Introduction (see also Methods), and ask the question of when do the results reported in Figs 1 and 2 actually signify an improvement in success and cooperation when compared with non-hybrid teams (see next Section). In Fig 3 the results for the substitution approach (see Methods) are provided wherein *a* of the non-hybrid group members are replaced by artificial agents. The expected gain in group success is calculated using Eq (15), where the observations as in Fig 1 are subtracted from the success of fully non-hybrid groups.

A general pattern can be identified where we observe that success is generally enhanced in low risk scenarios or in high risk scenarios where there are few agents ($N - a \geq a$, in panels A and B) and they are not very cooperative (low *p*). However, when risk is high and agents are cooperative, success appears to decrease in comparison with groups that would be made of $N = 6$ humans with the same threshold $M = 3$. For the special case of Fig 3C where $N - a < a$, when the agents are not cooperative and risk is high there is a big decrease in success when compared with groups made only of adaptive individuals, probably because the latter do not cooperate at all as we have observed in Fig 2C when grouped in hybrid teams of such a configuration.

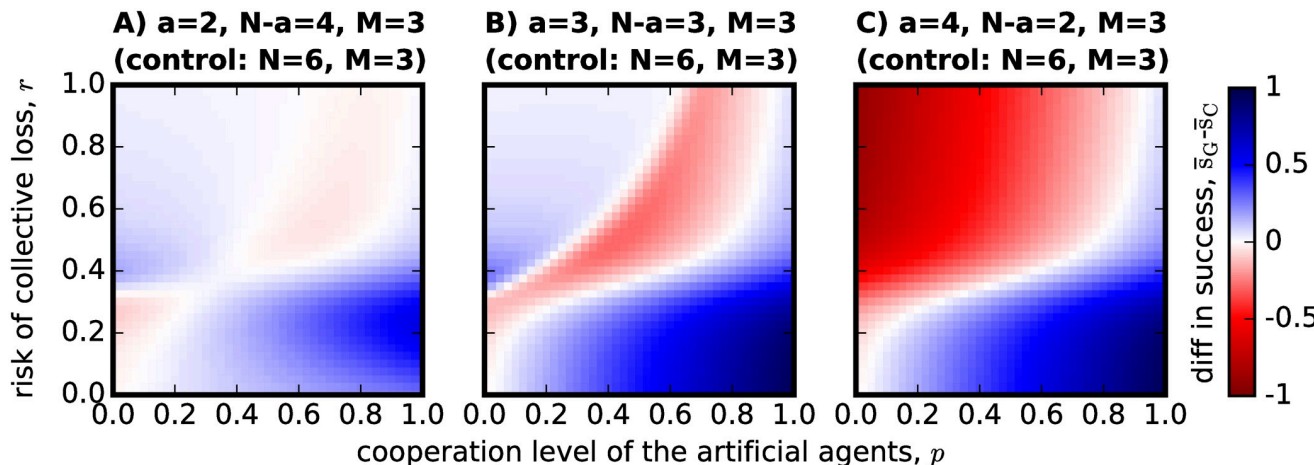

**Fig 3. The effect of *substitution*, as a process to form hybrid teams, on success rate.** The parameters referring to each hybrid group configuration are shown on top of each panel, as is the control parameters that allow us to calculate $\bar{s}_S$ (success rate for the non-hybrid groups before substitution). As in Fig 1: in (A) $N - a = 4$ and $a = 2$, in (B) $N - a = a = 3$ and in (C) $N - a = 2$ and $a = 4$. The plot shows that substituting part of the group by artificial agents with different cooperative levels only improves success when the risk probability is low. For high-risk CRD scenarios, substituting a high number of group members by artificial agents will greatly reduce the success rate that would otherwise be obtained by adaptive individuals playing the game alone. All images are produced following Eq (15) and use the same other parameters as the previous figures in this manuscript.

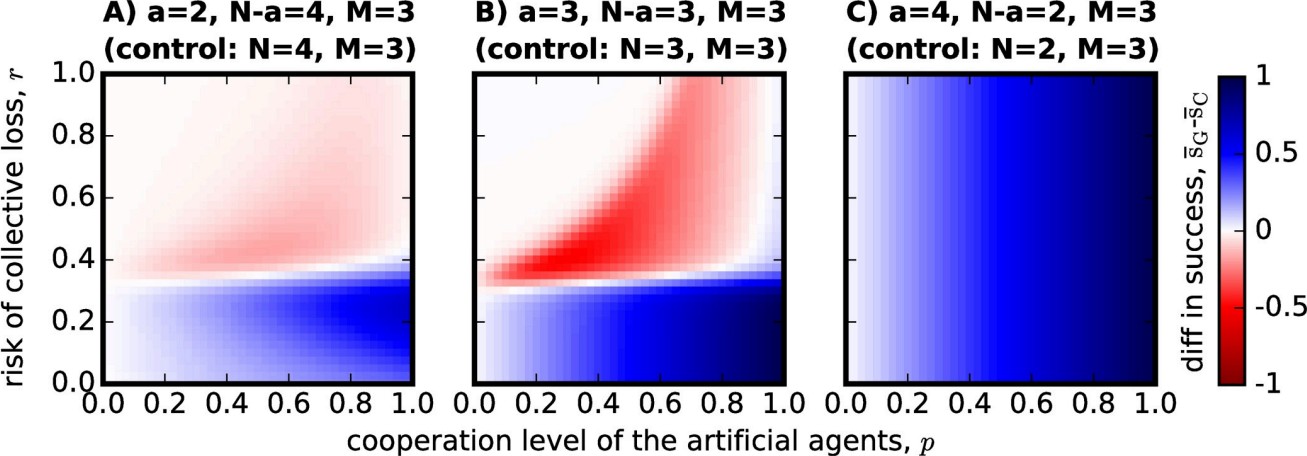

**Fig 4. The effect of *addition*, as a process to form hybrid teams, on success rate.** Again we re-use the same hybrid team configuration for each of the panels as in the previous figures. The control is now given by *addition*, changing the control parameters to the ones indicated in each panel title: in (A) the control group configuration consists of *N* = 4 adaptive agents, in (B) of *N* = 3 adaptive agents and in (C) of *N* = 2 adaptive agents. Note that unlike in Fig 3, the group configuration of the control in panel C is unable to meet the threshold without the added artificial agents. The results plotted follow Eq (17) and show that adding artificial agents to the group only improves success when the risk probability is low or when the adaptive individuals would not be able to reach the threshold *M* on their own (as in panel C where *N* − *a* < *M*). For high-risk CRD scenarios where *N* − *a* ≥ *M*, adding artificial agents will only reduce or, at best, maintain the success rate they would otherwise already achieve on their own.

Of course, substituting adaptive individuals by artificial agents is not the ideal application in a real-world scenario. AI ethics and the concerns for the future of work [41] indicate that AI agents (for which the artificial agents in this model try to account for) should be introduced in working teams to enhance the work that is already being performed, rather than to substitute human labour. With this in mind, we investigate the effect of *addition* (see Methods) as another process of forming hybrid teams. In Fig 4, we therefore compare the hybrid team configuration with a non-hybrid team made of the same number of $N_C$ adaptive agents, thus without the added *a* artificial agents.

Similarly to Fig 3, we observe that when the risk of collective disaster is low, adding artificial agents to the group will always benefit collective success. However, when the risk is high and we add a small number of cooperative agents to the group, this can hurt group achievement (Fig 4A and 4B). The main difference between substitution as shown in Fig 3 and addition as now plotted in Fig 4, as a mechanism to better group performance in the CRD, is that adding artificial agents to a group that would otherwise not be able to meet the threshold *M* will always increase the success of that group as long as the artificial agents are somewhat cooperative (*p* > 0). We also observe that for low *p* in Fig 4 there is no slight improvement of success in high risk scenarios, something that was also observed in Fig 3A and 3B, but a situation that we do not consider of relevance since it would be unlikely for an institution or organization to purposefully add non-cooperative agents to a hybrid team tackling a high-risk CRD.

## At high risk, highly cooperative artificial agents put the cooperative capacity of adaptive individuals at risk

Even though success in avoiding the risky outcome is an important metric when studying the CRD, we find it relevant to also understand how the introduction of artificial agents to form hybrid teams affects the evolution of cooperation. As previously mentioned in the introduction, the world is growing more dependent on the use of AI applications to support human

decision-making and performance, though we still have little understanding of how this machine dependence might affect human skills in the long-term. Limiting our study to social dilemma situations, the skill in question is precisely the human capacity for cooperation. Fig 5 presents what the effect of forming teams through a process of *substitution* might have on the average cooperation level obtained by the population of adaptive individuals. The figure shows again the results for the 3 different cases, as was done in Fig 3, now following Eq (14) to calculate the gain in cooperation.

Examining Fig 5, we can see that cooperation can be enhanced by the constitution of hybrid teams through the substitution of some group members by artificial agents, but only if certain conditions are met. When risk is high, we see that cooperation can be enhanced when the artificial agents in the group are not so cooperative (low $p$) and there are still at least as many humans in the group as the number of cooperators needed to meet the threshold ($N - a \geq M$), as in panels A and B of Fig 5. However, when the artificial agents are cooperative (high $p$), risk is high (high $r$)—reference to panels A and B—or the number of adaptive agents in the group would not suffice the cooperation requirements to achieve success on their own ($N - a < M$) —panel C -, then the adaptive population would exhibit higher levels of cooperation if not playing the CRD in a hybrid team. Again, following the analysis of Figs 3 and 4, we find that when the risk of collective disaster is low enough, substituting adaptive individuals by artificial agents has essentially no effect on cooperation (the exception being the small boost found in the case of $N - a > M > a$ in Fig 5A).

As previously discussed however, the hybrid team configuration might spur from the addition of artificial agents to a team made of $N - a$ adaptive agents, instead of substituting $a$ adaptive agents from a team of $N$. To study this effect of *addition* in the forming of hybrid teams, we produce Fig 6, where we use Eq (16) to plot the difference between the results already shown in Fig 2 and the results that an adaptive population would have obtained if playing in teams made only of $N - a = 4$ (panel A), $N - a = 3$ (panel B) and $N - a = 2$ (panel C) with $a = 0$.

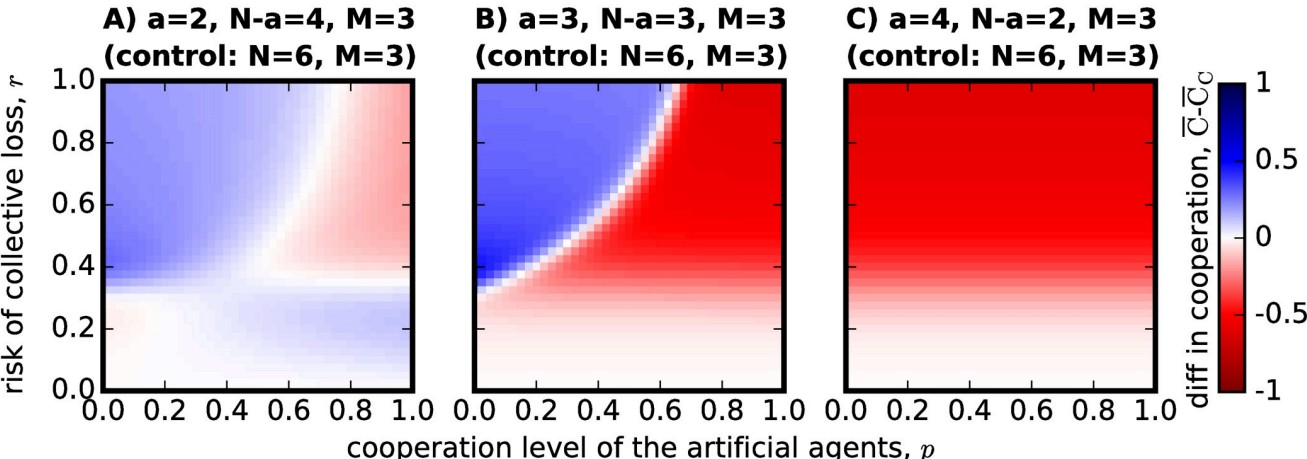

**Fig 5. The effect of *substitution*, as a process to form hybrid teams, on cooperation level.** All images were produced using the same parameters as the ones in Fig 3, now following Eq (14) to calculate the gain in cooperation level, rather than the effect on success rate. As indicated in the titles of each panel, the hybrid team configuration once again refers to: in (A) $N - a = 4$ and $a = 2$, in (B) $N - a = a = 3$ and in (C) $N - a = 2$ and $a = 4$. The plot shows that substituting part of the group by cooperative artificial agents will generally decrease the cooperation exhibited by the population if it was grouped in non-hybrid groups of the same group size $N$. The exception being the case of panel A, where $N - a > M > a$ for the region of low-risk. Substituting part of the group by less cooperative artificial agents might increase cooperation in the adaptive population when $N - a \geq a$ and risk is high, although it is an unlikely scenario in real-life applications for these to be introduced in high-risk CRD scenarios.

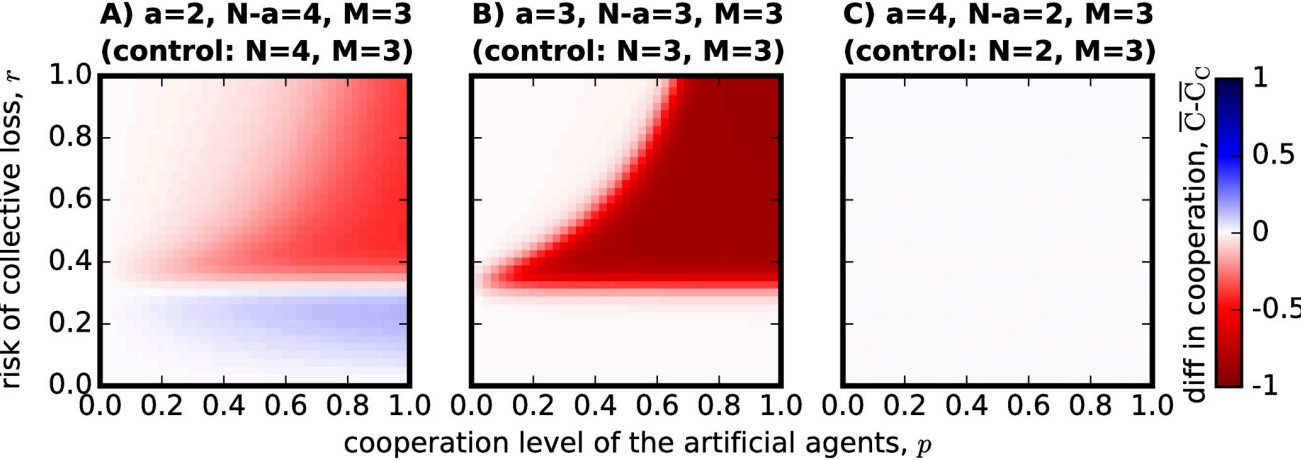

**Fig 6. The effect of *addition*, as a process to form hybrid teams, on cooperation level.** To reproduce this figure, the same parameters already used in Fig 4 are used as input into Eq (16). Therefore, as indicated in the subtitles of each panel: in (A) there are $N − a = 4$ adaptive agents in both the *control* and the hybrid case, with $a = 0$ in the control and $a = 2$ otherwise; in (B) $N − a = a = 3$ for the hybrid teams and $N − a = 3$ but $a = 0$ in the *control*; and in (C) $N − a = 2$ for both scenarios, but $a = 4$ in the hybrid case and $a = 0$ in the *control*. The plot shows that adding artificial agents to the group can only boost cooperation in the adaptive population when the risk $r$ is low and $N − a > a$. Otherwise, their addition will only decrease or the level of cooperation observed for a population where groups of the same $N − a$ number of adaptive individuals try to tackle the CRD on their own. When the risk is high and $N − a \geq a$, adding $a$ cooperative artificial agents to the group can greatly impair the cooperation level observed in the adaptive population, as shown in panels A and B.

Fig 6 further restricts the positive impact that hybrid teams might have in the evolution of cooperation on the adaptive population: only when the risk is low (low $r$) and the artificial agents added to the team could not reach the threshold on their own ($a < M$)—focus on Fig 6A -, does the addition of artificial agents benefit the cooperation capacity of the adaptive population. Whereas for low $p$ and high risk $r$ there was also a positive impact in the cooperation capacity when addressing the case of *substitution* in Fig 5, in this case of *addition* no difference is observed within this set of parameters. Similarly, even though substituting the greater part of a team by artificial agents would seriously impair that team's cooperative capacity when the risk is high (Fig 5C), adding artificial agents to a team that would not be able to meet the threshold on their own ($N − a < M$) does not change the cooperation level of that cooperative capacity (Fig 6C).

## Conclusions

In this work we investigated what behavior is selected by social learning in the context of the one-shot CRD, when interactions occur in hybrid groups made of adapting individuals (a proxy for human decision-making) and artificial agents with fixed probabilistic behaviors (a proxy for average AI behavior). This model is used as a thought experiment to reason about the behavior one could expect in hybrid groups of humans and AI agents. It focuses on mixed-motive situations where there is a conflict between individual and common interests, but also a risky outcome where the lack of cooperation does not immediately correspond to collective loss. One of the main contributions of this work is to reveal the importance of context and choice of the control group: even though it appears in hindsight as an obvious result, the addition of artificial agents to a group that would not be able to reach the threshold on its own always increases success (Fig 4C). However, the wrong choice of control, for example, if one would compare the hybrid team with a non-hybrid team of the same group size, its

performance appears to be very negative both in terms of group success in the high-risk region (Fig 3C) and in terms of the cooperation level (Fig 5C). To our knowledge, this work is the first concerned with this discussion on how the context of the hybrid team formation may influence its relative performance.

The model allowed us to understand that the benefits of introducing artificial agents in teams to help solve a CRD highly depends on the risk associated with the failure to do so: when the risk of loss is low, the introduction of artificial agents in the group always increases success; when it is high, this is not true. We find this model of special interest since most social dilemmas faced by our society also have a probabilistic, rather than deterministic, nature (for example, dealing with pandemics [3] or climate change [4]). Even so, humans will rely on biases and heuristics to evaluate risk, and will often misinterpret it [42]. Within this context, the proposed model is ever more relevant as it shows that an under- or overestimation of the risk of collective loss may lead to a wrong decision of whether or not to form hybrid teams to tackle the dilemma.

Moreover, with the model, we have disentangled how the changes in success rate are related to the effort produced by the artificial agents: whenever the latter are perceived as highly cooperative, the behavior of the adaptive population evolves to exploit the artificial agents' benevolence. An effect that has also been shown experimentally in [11]. However, when the artificial agents are low contributors, the adaptive population shifts to compensate those low contributions. Yet, this is only true for higher risk levels, as in this case, reaching the goal is the only means to secure payoff. These results are supported experimentally: in [16], hybrid teams were not significantly more successful than only-human teams, but humans were observed to contribute less at the expense of the artificial agents in their groups in a high-risk collective-risk dilemma. On the other hand, when risk is low and there is not enough incentive for the adaptive population to change their behavior, the addition of artificial agents to the group does seem to increase the group success rate. If the hybrid team does not solely depend on the artificial agents to meet the threshold, their addition might even contribute to a spike in the adaptive population in the long-term.

Even though previous experimental work [11, 16] corroborates some of the results reported here, additional behavioral experiments are needed to truly validate all the conclusions. Such experiments are left for future research given that a thorough experiment design with multiple controls (in this case, by using different human-only group sizes to test the hypothesis of addition vs substitution) would require a significant effort and resources.

In general, the discussed research suggests that there is potential benefit of using AI to increase the success rates of human-AI groups working together in low-risk scenarios. Nevertheless, our findings clearly show that the addition of artificial agents will not always be beneficial if the AI designer is also concerned with at least maintaining the cooperation level of the adaptive individuals before the hybrid team was formed. By working alongside cooperative AI, humans might eventually adapt to relax their own cooperative efforts. Hence, we must either identify AI policies that avoid this scenario and still promote cooperation to avoid collective risks or promote other modes of interaction in-between hybrid teams. Our model provides a tool to explore different risk scenarios simulated with different compositions of hybrid teams. Such a tool could become the first step in assessing how to safely constitute hybrid teams to effectively solve CRD-like problems.

## Author Contributions

**Conceptualization:** Inês Terrucha, Elias Fernández Domingos, Francisco C. Santos, Tom Lenaerts.

**Formal analysis:** Inês Terrucha.

**Funding acquisition:** Pieter Simoens, Tom Lenaerts.

**Investigation:** Inês Terrucha.

**Methodology:** Inês Terrucha.

**Project administration:** Tom Lenaerts.

**Software:** Inês Terrucha.

**Supervision:** Elias Fernández Domingos, Francisco C. Santos, Pieter Simoens, Tom Lenaerts.

**Validation:** Elias Fernández Domingos.

**Visualization:** Inês Terrucha.

**Writing – original draft:** Inês Terrucha.

**Writing – review & editing:** Inês Terrucha, Elias Fernández Domingos, Francisco C. Santos, Pieter Simoens, Tom Lenaerts.

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
