## [Decision Letter · Decision Letter 0]

19 Dec 2023

PONE-D-23-33002The art of compensation: how hybrid teams solve collective-risk dilemmasPLOS ONE

Dear Dr. Terrucha,

Thank you for submitting your manuscript to PLOS ONE. After careful consideration, we feel that it has merit but does not fully meet PLOS ONE’s publication criteria as it currently stands. Therefore, we invite you to submit a revised version of the manuscript that addresses the points raised during the review process.

**ACADEMIC EDITOR: **The received feedback from the referee and the associate editor is that the paper is well written and addresses an important and timely problem. It would make an excellent contribution to the theoretical literature of the evolution of cooperation and provides useful insights for the design of prosocial AI. The reviewer has provided constructive suggestions -- please carefully take them into account when preparing the revised version.

We look forward to receiving your revised manuscript.

Kind regards,

The Anh Han, Ph.D.

Academic Editor

PLOS ONE

“I.T., P.S. and T.L are supported by an FWO (Fond Wetenschappelijk Onderzoek) project with grant no. G054919N. E.F.D. is supported by an F.R.S-FNRS (Fonds de la Recherche Scientifique) Chargé de Recherche grant (nr. 40005955). T.L. is furthermore supported by two F.R.S.-FNRS PDR (grant numbers 31257234 and 40007793) and acknowledges also the support by TAILOR, a project funded by EU Horizon 2020 research and innovation programme under GA No 952215.. E.F.D. and T.L. are supported by Service Public de Wallonie Recherche under grant n° 2010235–ariac by digitalwallonia4.ai. T.L. and P.S. acknowledge the support by the Flemish Government through the Flanders AI Research Program.”

“I.T., P.S. and T.L are supported by an FWO (Fond Wetenschappelijk Onderzoek) project with grant no. G054919N. E.F.D. is supported by an F.R.S-FNRS (Fonds de la Recherche Scientifique) Charg´e de Recherche grant (nr. 40005955). T.L. is furthermore supported by two F.R.S.-FNRS PDR (grant numbers 31257234 and 40007793) and acknowledges also the support by TAILOR, a project funded by EU Horizon 2020 research and innovation programme under GA No 952215. E.F.D. and T.L. are supported by Service Public de Wallonie Recherche under grant n° 2010235–ariac by digitalwallonia4.ai. T.L. and P.S. acknowledge the support by the Flemish Government through the Flanders AI Research Program.”

We note that you have provided additional information within the Acknowledgements Section that is currently declared in your Funding Statement. Please note that funding information should not appear in the Acknowledgments section or other areas of your manuscript. We will only publish funding information present in the Funding Statement section of the online submission form.

“I.T., P.S. and T.L are supported by an FWO (Fond Wetenschappelijk Onderzoek) project with grant no. G054919N. E.F.D. is supported by an F.R.S-FNRS (Fonds de la Recherche Scientifique) Chargé de Recherche grant (nr. 40005955). T.L. is furthermore supported by two F.R.S.-FNRS PDR (grant numbers 31257234 and 40007793) and acknowledges also the support by TAILOR, a project funded by EU Horizon 2020 research and innovation programme under GA No 952215.. E.F.D. and T.L. are supported by Service Public de Wallonie Recherche under grant n° 2010235–ariac by digitalwallonia4.ai. T.L. and P.S. acknowledge the support by the Flemish Government through the Flanders AI Research Program.”

4. We are unable to open your Supporting Information file [supporting information.ipynb]. Please kindly revise as necessary and re-upload.

Additional Editor Comments:

The received feedback from the referee and the associate editor is that the paper is well written and addresses an important and timely problem. It would make an excellent contribution to the theoretical literature of the evolution of cooperation and provides useful insights for the design of prosocial AI. The reviewer has provided constructive suggestions -- please carefully take them into account when preparing the revised version.

Reviewers' comments:

Reviewer's Responses to Questions

**Comments to the Author**

1. Is the manuscript technically sound, and do the data support the conclusions?

Reviewer #1: Yes

2. Has the statistical analysis been performed appropriately and rigorously? 

Reviewer #1: N/A

3. Have the authors made all data underlying the findings in their manuscript fully available?

Reviewer #1: Yes

4. Is the manuscript presented in an intelligible fashion and written in standard English?

Reviewer #1: Yes

5. Review Comments to the Author

Reviewer #1: In this manuscript, the authors developed an agent-based model where two types of agents play the one-shot collective risk dilemma (CRD). One type is supposed to be a human-like agent. Those agents optimize their strategies for the CRD based on the framework of evolutionary games. The other type is supposed to be AI-like agents. Those agents cooperate in the CRD with probability p. Using this model, the authors find that interesting phenomena shown in Fig. 1B and 2B. When the AI-like agents cooperate with high probability (bottom right area in Fig. 1B), the human-like agents do not cooperate. They free ride on the cooperation of the AI-like agents. On the other hand, when the AI-like agents cooperate with low probability (top left area in in Fig. 1B), the human-like agents actively cooperate to accomplish the goal of the CRD. The authors argued that those two types of agents compensate with each other. Thus, these human-AI hybrid teams works well to prevent the collective risks. Then, the authors considered the situation how the hybrid team works well by substitution or addition of AI-like agents to human-like agents. The model is new and worth investigating from the view points of not only evolutionary games but also human-AI interaction. I suggest two minor comments below. Please take them into account for improving the manuscript more.

1. Obviously, readers would like to know whether real humans and real AI (AI bots) actually compensate with each other when playing the CRD. It can be tested by conducting online experiments. But this is out of the scope of this manuscript because the authors take a model-based approach here. However, it is worth mentioning this point in the Conclusions by using one paragraph. If the authors agree with my suggestion, please do so.

2. It is worth citing the following royal society interface paper because they also simulated human-AI evolutionary games based on a model-based approach.

Small bots, big impact: solving the conundrum of cooperation in optional Prisoner’s Dilemma game through simple strategies, Gopal Sharma, Hao Guo, Chen Shen and Jun Tanimoto

Typos:

1. Ref. [32] is double cited on lines 127 and 258.

2. line 280: with the results in 1  with the results in 1A

6. PLOS authors have the option to publish the peer review history of their article (what does this mean?). If published, this will include your full peer review and any attached files.

Reviewer #1: No

---

## [Author Response · Author response to Decision Letter 0]

22 Dec 2023

We thank the reviewer for the positive assessment of our work. Please find below the answers to your two remaining comments. 

1. Thank you for your comment, we absolutely agree with the importance of validating our theoretical results with behavioral experiments and have added this remark as a paragraph to our Conclusions. The new paragraph should appear highlighted in the document 'Revised Manuscript with Track Changes' in lines 453-458.

2. Thank you for making us aware of this reference, it is indeed of great interest to our work and we have added it in the Related Work section in line 122 as the new reference [32], as highlighted in the document 'Revised Manuscript with Track Changes'.

And finally, we gladly confirm that the typos found were corrected, and we thank the reviewer again for their time and attention in revising our manuscript and helping us improve our work.

---

## [Editor Report · Decision Letter 1]

2 Jan 2024

The art of compensation: how hybrid teams solve collective-risk dilemmas

PONE-D-23-33002R1

Dear Dr. Terrucha,

We’re pleased to inform you that your manuscript has been judged scientifically suitable for publication and will be formally accepted for publication once it meets all outstanding technical requirements.

Kind regards,

The Anh Han, Ph.D.

Academic Editor

PLOS ONE

---

## [Editor Report · Acceptance letter]

31 Jan 2024

PONE-D-23-33002R1 

PLOS ONE

Dear Dr. Terrucha, 

I'm pleased to inform you that your manuscript has been deemed suitable for publication in PLOS ONE. Congratulations! Your manuscript is now being handed over to our production team.

Kind regards, 

on behalf of

Dr. The Anh Han 

Academic Editor

PLOS ONE